# GSPLANE: CONCISE AND ACCURATE PLANAR RE-CONSTRUCTION VIA STRUCTURED REPRESENTATION

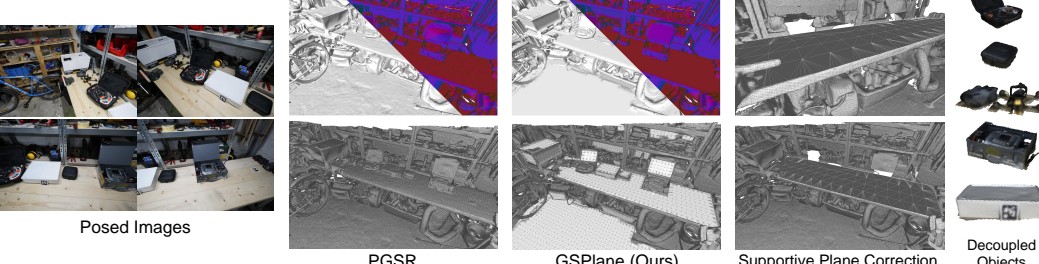

Figure 1: We introduce GSPlane, which adopts 2D planar priors to constrain planar Gaussian distributions on corresponding planes. The structured representation for planes not only empowers the layout refinement of the mesh, resulting in topological correctness with notable less vertices, but also demonstrates the potential in decoupling objects and sealing contact regions on supportive planes.

## ABSTRACT

Planes are fundamental primitives of 3D sences, especially in man-made environments such as indoor spaces and urban streets. Representing these planes in a structured and parameterized format facilitates scene editing and physical simulations in downstream applications. Recently, Gaussian Splatting (GS) has demonstrated remarkable effectiveness in the Novel View Synthesis task, with extensions showing great potential in accurate surface reconstruction. However, even state-of-the-art GS representations often struggle to reconstruct planar regions with sufficient smoothness and precision. To address this issue, we propose **GSPlane**, which recovers accurate geometry and produces clean and well-structured mesh connectivity for plane regions in the reconstructed scene. By leveraging off-the-shelf segmentation and normal prediction models, GSPlane extracts robust planar priors to establish structured representations for planar Gaussian coordinates, which help guide the training process by enforcing geometric consistency. To further enhance training robustness, a **Dynamic Gaussian Re-classifier** is introduced to adaptively reclassify planar Gaussians with persistently high gradients as non-planar, ensuring more reliable optimization. Furthermore, we utilize the optimized planar priors to refine the mesh layouts, significantly improving topological structure while reducing the number of vertices and faces. We also explore applications of the structured planar representation, which enable decoupling and flexible manipulation of objects on supportive planes. Extensive experiments demonstrate that, with no sacrifice in rendering quality, the introduction of planar priors significantly improves the geometric accuracy of the extracted meshes across various baselines.

## 1 INTRODUCTION

Planes are commonly witnessed in our daily environments, forming the foundation of many scenes: the streets and building facades outdoors, and the floors and ceilings indoors. When manually constructing digital assets, artists can easily leverage their priors knowledge to accurately model textures and geometric distributions in these areas. Establishing accurate planar structures not only enables concise meshes with considerably fewer vertices and faces, but also support downstream tasks such

as physical simulation Qi et al. (2024). In contrast, for example, if a reconstructed table is uneven or lacks geometric consistency, objects like cups would struggle to rest stably on its surface. Naturally, a question arises: in the context of creating digital twins through 3D reconstruction, can the introduction of planar priors help achieve more normal-consistent and accurate geometric reconstructions? Unfortunately, despite recent advancements in 3D reconstruction, there has been limited exploration of how to effectively leverage planar priors to address these challenges.

Recently, 3D Gaussian Splatting (3DGS) Kerbl et al. (2023) introduced an explicit representation capable of achieving high-fidelity novel view synthesis in real time. Rather than relying on neural networks, 3DGS employs Gaussians characterized by parameters such as position, scale, rotation, color, and opacity. Its highly optimized rasterization pipeline enables fast rendering speed. Following 3DGS, several notable works Huang et al. (2024a); Guédon & Lepetit (2024); Zhang et al. (2024); Yu et al. (2024); Chen et al. (2024) focused on improving the Gaussian representation and depth regularization strategies to gain higher mesh quality in 3D surface reconstruction tasks, which has been extensively studied in the field of computer vision and graphics. Building on these advancements, methods such as GaussianRoom Xiang et al. (2024), AGS-mesh Ren et al. (2024) further integrate prior information to enhance geometric accuracy. However, through our experiments, we observed that the prior knowledge in these methods is typically used as a supervisory signal to minimize regularization losses during training, and the Gaussian representations generated are not strictly constrained to lie on a single plane. Additionally, the meshing strategies adopted in these approaches tend to produce overly dense distributions of vertices and faces, especially for planar regions, leading to high-resolution demands that can be costly and less practical for downstream applications.

To address the aforementioned challenges, we propose **GSPlane**, a novel method that leverages planar priors from 2D images to generate meshes with consistent normal and coherent topology in planar regions. Our approach begins by estimating surface normal maps Hu et al. (2024) for each posed image and identifying potential planar regions using subpart mask proposals generated by SAM Kirillov et al. (2023). These 2D planar priors are then projected into 3D space to cluster the initial 3D Gaussians into plane-specific groups. We introduce a structured representation for planar Gaussians by re-parameterizing their $xyz$ coordinates into a normalized weighted combination of three non-collinear basis points defining the plane. During training, both the basis points' coordinates and the normalized weights for each planar Gaussian are optimized to refine the plane's orientation and position. To further improve accuracy, we incorporate a **Dynamic Gaussian Reclassifier** (DGR), which dynamically corrects false-positive planar Gaussians during training. The extracted mesh will be further refined by leveraging the optimized planar priors, enhancing the surface topology and layout in planar regions. Additionally, we explore **Supportive Plane Correction** (SPC), an applications of our structured planar representation, demonstrating its ability to improve mesh realism by preserving planar integrity and enabling flexible object manipulation across supportive planes.

To thoroughly evaluate the effectiveness of 2D planar priors, we take both the indoor dataset ScanNetV2 Dai et al. (2017) and outdoor Tanks and Temples Dataset Knapitsch et al. (2017) as benchmarks. Extensive experiments demonstrate that GSPlane achieves significantly better performance in planar regions, producing meshes with a unified layout and consistent normals—while maintaining rendering quality without any degradation. To summarize, the main contributions of the paper are:

- We propose **GSPlane**, a powerful method that lifts 2D planar priors into 3D space and establishes a structured representation for planar Gaussians. Additionally, we incorporate optimized planar information during mesh layout refinement, ensuring topological correctness and consistency in the planar regions of the mesh.

- We present Supportive Plane Correction, an application of our structured planar representation that preserves planar integrity when decoupling objects from their supportive planes, enabling accurate planar geometry and facilitating flexible object manipulation.

- Extensive experiments validate our SOTA surface reconstruction performance, showcasing promising benefits of 2D planar prior in 3D reconstruction.

## 2 RELATED WORKS

### 2.1 GAUSSIAN SPLATTING

Extracting accurate surfaces from unordered and discrete 3DGS is both a challenging and fascinating task. Numerous algorithms have been developed to extract high-quality surfaces while ensuring smoothness and managing outliers. The pioneering SuGaR Guédon & Lepetit (2024) approach pretrains 3DGS and integrates it with the extracted mesh for fine-tuning, utilizing the Poisson reconstruction algorithm for rapid mesh extraction. Techniques like 2DGS Huang et al. (2024a) and GaussianSurfels Dai et al. (2024) reduce the original 3D Gaussian primitives to 2D to avoid ambiguous depth estimation. During GS training, the estimated normals derived from rendering and depth maps are aligned to ensure smooth surfaces. GOF Yu et al. (2024) focuses on unbounded scenes, using ray-tracing-based volume rendering to achieve a contiguous opacity distribution. RaDeGS Zhang et al. (2024) introduces a novel definition of ray intersection with Gaussian structures, deriving curved surfaces and depth distributions. Furthermore, recent works Xiang et al. (2024); Ren et al. (2024); Turkulainen et al. (2024); Wang et al. (2024); Dai et al. (2024); Chen et al. (2024); Zanjani et al. (2025); Li et al. (2025); Sun et al. (2025) incorporate surface normal and monocular depth information predicted from off-the-shelf models as additional supervision in the training process, resulting in improved surface reconstruction quality and geometrical consistency. However, these mesh surfaces are still composed of overly dense distributions of vertices and faces, resulting in topological inaccuracies when compared to real-world structures. This excessive density not only leads to significantly larger file sizes but also poses challenges for subsequent editing and processing tasks.

### 2.2 TRADITIONAL 3D PLANE RECONSTRUCTION

Traditional methods for 3D plane reconstruction often focus on identifying potential plane areas within a scene using RGB-D images Salas-Moreno et al. (2014); Silberman et al. (2012); Huang et al. (2017) or sparse 3D point clouds Borrmann et al. (2011); Sommer et al. (2020). By utilizing sets of points with 3D coordinates, either obtained from point clouds or derived from depth information, robust estimators such as PCA or RANSAC Fischler & Bolles (1981) can be employed to fit geometric representations of planes. Other approaches Gallup et al. (2010); Argiles et al. (2011) tackle the planar reconstruction problem through multi-view image segmentation, where each pixel is assigned to planar proposals represented in Markov Random Fields (MRF). In our research, we propose leveraging planar priors from 2D images to reconstruct target scenes. In earlier attempts, we proposed to directly post-process the reconstructed mesh Barda et al. (2023) via 2D planar priors, which led to significant errors in plane distribution. To address this, we introduced a structured planar representation that is optimized during training, allowing us to leverage learned plane equations to refine the reconstruction.

### 2.3 LEARNABLE 3D PLANE RECONSTRUCTION

With the increasing availability of large-scale datasets containing both 2D images and 3D point clouds, learning-based methods have become the mainstream for extracting planar information from single images or videos. This capability facilitates the reconstruction of potential planes within a scene. Classical approaches, such as PlaneNet Liu et al. (2018), PlaneRecover Yang & Zhou (2018), and PlaneRCNN Liu et al. (2019), segment possible plane distributions from a single image and optimize plane parameters using depth features to achieve a final reconstructed scene. PlanarRecon Xie et al. (2022) is the first method to predict the planar representation of a scene from a sequence of images before reconstruction. Building on previous methods, Airplanes Watson et al. (2024) proposes estimating 3D-consistent plane embeddings and grouping them into scene instances. Uniplane Huang et al. (2024b) uses sparse attention to query per-object embeddings for the scene. Alphatablets He et al. (2024) employs off-the-shelf surface normal and depth information to initialize small planes, which are further optimized to align with the scene's geometry and texture. While these methods show significant promise in reconstructing planar regions, they often produce less detailed and realistic geometric structures in non-planar areas. In contrast, out model well balance the performance in both planar and non-planar areas, achieving high quality for both rendering and surface reconstruction.

162
163
164

# 3 METHODS

165
166
167
168
169
170
171
172
173

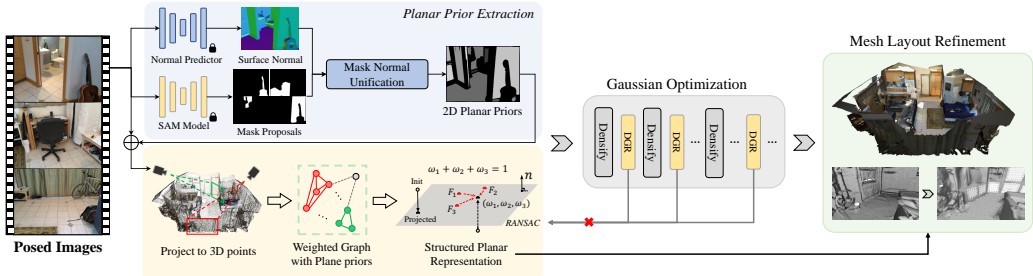

174
175
176
177
178
179

Figure 2: Pipeline of **GSPlane**. Given a set of posed images as input, our method first extracts 2D planar priors from each view, align them with point cloud to obtain plane distributions in 3D space, and re-parameterize the related coordinates of Gaussians. During training, our Dynamic Gaussian Re-classifier continues to correct false-positive planar gaussian by reverting their representation back to $xyz$. The layout of the mesh extracted from training will also be refined with the optimized planar information from the structured representation.

180
181
182
183
184
185
186
187
188

Figure 2 illustrates the overall pipeline of **GSPlane**. Starting with posed input images, **GSPlane** initially extracts planar prior information from each specific view, integrates them into the 3D point cloud, and establishes structured representations for 3D planar points before and during training (Sec. 3.1). A **Dynamic Gaussian Re-classifier** (DGR) is then employed to refine the optimization process by identifying and correcting false-positive planar Gaussians (Sec. 3.2). Finally, the extracted mesh is refined using the learned planar distributions to enhance surface topology and layout (Sec. 3.3). Additionally, we propose **Supportive Plane Correction** (SPC), an application incubated from planar prior to improve realism by preserving planar integrity and enabling flexible object manipulation in reconstructed scenes (Sec. 3.4).

189
190

## 3.1 STRUCTURED REPRESENTATION FOR PLANES

191
192
193
194
195
196
197
198
199

Given a set of posed images $I_p = \{I_1, I_2, \ldots, I_n\}$, potential planes are detected in each image using surface normal predictions. For each image $I_i$, Metric3Dv2 Hu et al. (2024) generates a surface normal map $N_i = (n_x, n_y, n_z) \in \mathbb{R}^{H \times W \times 3}$ and Segment-Anything-Model (SAM) Kirillov et al. (2023) produces subpart masks $M_i = \{M_{i,1}, M_{i,2}, \ldots, M_{i,j}\}$ of the scene, where $i$ denotes the $i$-th image and $j$ for $j$-th mask. For each mask region $M_{i,j}$, we compute the cosine similarity between the normals of individual pixels and the average normal of the region. If more than 70% of the pixels in the region exceed the similarity threshold $\alpha$, these pixels are identified as a planar region. Overlapping planar regions are then merged into larger planar masks $P = \{P_1, P_2, ..., P_n\}$ due to normal consistency.

200
201
202
203
204
205
206
207
208
209
210

Given an initial point cloud or COLMAP reconstruction of the scene, the coordinates of points in the point cloud are used to initialize the positions of the Gaussians. To incorporate planar priors into training, we establish planar relationships across different Gaussian units by projecting 2D planar masks from multiple views into 3D space. we then construct an undirected graph $G(V, E)$, where each node $V_i$ corresponds to a point in the point cloud. An edge $E(V_i, V_j)$ is established between two nodes if the two corresponding points appear together on the same projected planar mask. The weight of the edge represents the frequency of these two points appearing on the same planar mask. Background points can be filtered out using depth information, and planar relationships are aggregated across all views on the graph $G$. The Leiden algorithm Traag et al. (2019), which is designed to detect communities in weighted graph, clusters nodes in $G$ into planar groups, and will be served as constraints for training Gaussians. More details can be found in Appendix Sec. A.

211
212
213
214
215

We assume that if a group of points $V_P$ in the point cloud is determined to lie on a plane, their corresponding Gaussian centers should also reside on the same plane. To impose this constraint, we introduce planar priors to re-parameterize Gaussian coordinates, replacing the direct optimization of $xyz$ positions with normalized weight parameters. Specifically, for a planar cluster $V_P$, RANSAC Fischler & Bolles (1981) is employed to estimate the plane, onto which all Gaussian centers are projected to obtain $V_P'$. From $V_P'$, three non-collinear points $F_1, F_2, F_3$ are randomly

selected to serve as basis points defining the plane function. The sample follows the restriction that the distance between each point should exceed 1/4 planar area diameter to prevent rapid rotation of the plane. Each projected coordinate in $V_P'$ is then expressed as a normalized linear combination of the basis points:

$$V_P' = \omega_1 F_1 + \omega_2 F_2 + \omega_3 F_3, \quad \text{s.t.} \quad \omega_1 + \omega_2 + \omega_3 = 1. \tag{1}$$

These weights $\omega_1, \omega_2, \omega_3$ are optimized during training to enforce planar constraints on the planar Gaussians.

### 3.2 DYNAMIC GAUSSAIN RE-CLASSIFIER

Building upon the structured representation, planar Gaussians are optimized during training to adhere to planar constraints. The coordinates of planar Gaussians, whether initialized directly from the point cloud or derived through densification, are represented using basis points and normalized weights (Eq. 1). While the coordinates of the basis points are optimized as well, they are assigned a lower learning rate to allow for adjustments in plane orientation and position.

The accuracy of the planar Gaussian relations and the effectiveness of the planar priors are closely tied to the performance of SAM and Metric3Dv2. However, in cases where a Gaussian is misclassified as planar (*i.e.*, a false-positive planar Gaussian), it cannot be correctly optimized according to the planar coordinate formulation in Eq. 1. To address this issue, we propose the **Dynamic Gaussian Re-classifier** (DGR) to identify and reclassify such false-positive planar Gaussians. During the DGR phases, gradients for both planar and non-planar Gaussians are collected and averaged for evaluation. The top 5% of planar Gaussians, based on their average gradients, are then compared to the average gradient magnitude of the top 20% of non-planar Gaussians. If the gradient magnitude of a planar Gaussian exceeds the average gradient magnitude of the top 20% non-planar Gaussians, the coordinates of that planar Gaussian are re-formulated back into the $xyz$ coordinate format. DGR operates iteratively between Gaussian densification and after the final densification step. The implementation details are provided in Sec. B in the Appendix.

### 3.3 MESH LAYOUT REFINEMENT

Traditional mesh generation methods applied after Gaussian Splatting often produce overly dense meshes with redundant vertices and faces, which not only reduce geometric accuracy but also compromise storage efficiency. To address this, we introduce a mesh layout refinement procedure that leverages planar priors to optimize mesh structure in planar regions. This refinement improves normal consistency, topological coherence, and reduces vertex density, facilitating object decoupling from supportive planes like floors or tables.

Starting from an initial mesh $O$ (e.g., generated via TSDF Curless & Levoy (1996), Marching Tetrahedra Shen et al. (2021), etc.), we first identify clusters of mesh vertices that correspond to known planar regions. These planar relationships are precomputed from the sparse point cloud $Pcd$ as sets $V_P^i$, where $i$ indexes the $i$-th detected plane. We transfer these planar relationships from the point cloud to the mesh by assigning mesh vertices to planes using a spatial proximity criterion based on the voxel size $\delta$. Specifically, for a given plane $A$, a mesh vertex $v_x \in O$ is considered to belong to plane $A$ if:

$$\{v_x \mid \exists v_y \in V_P^A, \ |v_y - v_x| < 1.5\delta \ \land \ \forall \bar{v}_y \notin V_P^A, \ |\bar{v}_y - v_x| > 0.5\delta\}. \tag{2}$$

This ensures that each mesh vertex is matched to a unique planar region with sufficient spatial confidence.

Once planar vertex clusters are identified, we refine each planar region individually. We begin by removing all mesh faces formed by three vertices lying on the same plane, retaining only the associated vertices. Each planar vertex cluster is then classified into two categories. **Boundary vertices**, which are connected via mesh edges to vertices outside the planar cluster and thus form the perimeter of the planar region. **Interior vertices**, which are fully enclosed within the planar region and have no direct connections to non-cluster vertices. Both boundary and interior vertices are projected onto their corresponding planes, defined by the optimized basis points. To regularize the interior structure, we replace interior vertices with a set of uniformly distributed 2D grid points on the plane. These grid points serve as candidates for reconstructing the triangulated surface of

the planar region. However, we observe that planar regions in meshes often have irregular shapes, which can cause misalignment between the grid layout and the actual geometry. To mitigate this, we compute the minimum enclosing rectangle (MER) of the projected vertices. The MER provides a consistent 2D bounding frame aligned with the local plane axes, enabling uniform placement of grid points along the $x$- and $y$-directions. Considering the actual region of the plane in mesh, grid points falling outside the projected planar region are discarded. The remaining grid points, together with the projected boundary vertices, form a 2D point setthat is triangulated using Delaunay triangulation Lee & Schachter (1980).This produces a set of triangular faces that seamlessly connect the planar interior to its boundary. Finally, the 2D grid coordinates and their associated faces are mapped back into 3D space using the plane basis, and the resulting vertices and faces are integrated into the original mesh. This results in a refined planar region with consistent normals, reduced redundancy, and improved geometric structure. The complete mesh refinement algorithm is detailed in Alg. 2 in the Appendix.

## 3.4 SUPPORTIVE PLANE CORRECTION

Conventional mesh reconstruction methods often merge individual objects and structural elements into a single, overly connected surface. This results in unrealistic geometry, particularly in regions where objects are in contact. For instance, when attempting to digitally separate an object - such as removing a cup from a table - the reconstructed mesh may exhibit gaps or voids in the contact area, failing to preserve the original physical continuity of the supporting surface. To address this challenge, we propose leveraging planar priors to refine mesh representations within designated planar regions. This approach, referred to as **Supportive Plane Correction** (SPC), is an optional refinement step in our method designed to handle planar surfaces that serve as object-supporting structures, such as tables, shelves, or floors. To address this issue, we introduce an optional refinement step termed **Supportive Plane Correction** (SPC), which leverages planar priors to improve mesh representations of object-supporting surfaces, such as tables, shelves, or floors. Unlike general planar regions, supportive planes typically exhibit structural incompleteness — characterized by multiple internal voids (e.g., holes within the plane) or missing boundary regions (e.g., incomplete edges). SPC builds upon the mesh layout refinement process described in Sec. 3.3, with key modifications tailored to preserve the integrity of supportive planes. Specifically, during grid point sampling, points that fall outside the initially projected planar region are *retained* rather than discarded. In contrast, boundary vertices that define voids or holes are *excluded* from the Delaunay triangulation step.This ensures that the resulting triangulated surface spans the full extent of the plane while avoiding reintroducing known discontinuities. Beyond structural refinement, SPC enables flexible and physically plausible object manipulation. By isolating and sealing the contact regions between objects and their supporting surfaces, individual objects can be repositioned or removed without affecting the geometry of the underlying plane. This capability enhances both the visual realism and editability of the reconstructed scene by preserving planar surface continuity while enabling object-level interaction.

## 4 EXPERIMENTS

### 4.1 EXPERIMENTAL SETTINGS

**Dataset.** We conduct extensive experiments on both the indoor dataset ScanNetV2 Dai et al. (2017) and outdoor dataset Tanks and Temples Dataset Knapitsch et al. (2017). Both datasets provides ground-truth mesh for evaluation. We evaluate scenes in terms of geometric accuracy, plane-wise geometric accuracy, and rendering quality compared with previous methods.

**Metrics.** To evaluate the **scene-wise** geometric reconstruction performance, we follow the protocol of PlanarRecon Xie et al. (2022) and report metrics including *Accuracy*, *Completion*, *Precision*, *Recall*, and *F-score*. Additionally, we adopt the approach from Airplanes Watson et al. (2024) to report **planar-wise** metrics such as *fidelity*, *completion*, and *L1 chamfer*. These metrics are evaluated on the $k = 20$ and $k = 30$ largest planes sampled from ground truth mesh using PlaneRCNN Liu et al. (2019). Note that planar-wise metrics can only be assessed on meshes produced through our Planar-Guided Mesh Extraction, as baseline methods do not incorporate planar information in the extracted mesh. Please refer to airplanes Watson et al. (2024) for more details. To comprehensively evaluate performance, we also provide metrics about rendering quality, including PSNR, SSIM, and LPIPS, as done in 3DGS Kerbl et al. (2023).

| Method | Geometry | | | | | NVS | | | Mesh |
| | Acc↓ | Comp↓ | Prec↑ | Recall↑ | F-score↑ | SSIM↑ | PSNR↑ | LPIPS↓ | Vertices |
|---|---|---|---|---|---|---|---|---|---|
| GaussianRoom Xiang et al. (2024) | 0.084 | 0.062 | 0.602 | 0.621 | 0.611 | 0.779 | 23.89 | 0.36 | 3.01M |
| Alphatablets He et al. (2024) | 0.094 | 0.219 | 0.501 | 0.446 | 0.459 | - | - | - | 139.4K |
| 3DGS Kerbl et al. (2023) | 0.083 | 0.099 | 0.453 | 0.429 | 0.436 | 0.849 | 23.494 | 0.321 | 2.24M |
| 3DGS + Ours-train | 0.088 | 0.097 | 0.459 | 0.438 | 0.446 | **0.853** | **23.718** | **0.320** | 2.00M |
| 3DGS + Ours-full | **0.077** | **0.080** | **0.471** | **0.656** | **0.548** | - | - | - | **1.23M** |
| 2DGS Huang et al. (2024a) | 0.066 | 0.078 | 0.603 | 0.568 | 0.583 | 0.845 | 22.673 | 0.346 | 1.73M |
| 2DGS + Ours-train | 0.063 | 0.073 | 0.650 | 0.620 | 0.633 | **0.847** | **23.263** | **0.337** | 1.60M |
| 2DGS + Ours-full | **0.058** | **0.062** | **0.664** | **0.716** | **0.689** | - | - | - | **946.1K** |
| GOF (Tetra.) Yu et al. (2024) | 0.120 | 0.111 | 0.413 | 0.484 | 0.444 | 0.810 | 21.444 | **0.357** | 41.7M |
| GOF (Tetra.) + Ours-train | 0.108 | 0.101 | 0.457 | 0.526 | 0.489 | **0.828** | **22.460** | 0.359 | 41.2M |
| GOF (Tetra.) + Ours-full | **0.095** | **0.092** | **0.472** | **0.579** | **0.520** | - | - | - | **38.9M** |
| GOF (TSDF) Yu et al. (2024) | 0.107 | 0.113 | 0.443 | 0.511 | 0.474 | 0.810 | 21.444 | **0.357** | 2.36M |
| GOF (TSDF) + Ours-train | 0.100 | 0.091 | 0.477 | 0.598 | 0.528 | **0.828** | **22.460** | 0.359 | 1.89M |
| GOF (TSDF) + Ours-full | **0.086** | **0.080** | **0.482** | **0.686** | **0.566** | - | - | - | **1.02M** |
| RaDe-GS Zhang et al. (2024) | 0.101 | 0.104 | 0.480 | 0.507 | 0.491 | 0.829 | 22.334 | **0.348** | 1.49M |
| RaDe-GS + Ours-train | 0.096 | 0.101 | 0.507 | 0.558 | 0.528 | **0.832** | **22.394** | 0.351 | 1.45M |
| RaDe-GS + Ours-full | **0.082** | **0.086** | **0.520** | **0.674** | **0.587** | - | - | - | **794.3K** |
| PGSR Chen et al. (2024) | 0.079 | 0.085 | 0.581 | 0.571 | 0.573 | 0.847 | 25.350 | 0.274 | 5.3M |
| PGSR + Ours-train | 0.065 | 0.063 | 0.633 | 0.640 | 0.634 | **0.852** | **25.494** | **0.261** | 5.2M |
| PGSR + Ours-full | **0.062** | **0.059** | **0.636** | **0.658** | **0.646** | - | - | - | **2.9M** |

Table 1: Quantitative evaluations including both the overall geometric scores and novel view synthesis (NVS) metrics on ScanNetV2 Dai et al. (2017) scenes. 'Ours-train' denotes applying structured representation for planes and DGR. 'Ours-full' denotes additionally applying mesh layout refinement after training.

| Metric | 3DGS | 3DGS + Ours | 2DGS | 2DGS + Ours | GOF | GOF + Ours | RaDe-GS | RaDe-GS + Ours | PGSR | PGSR + Ours |
|---|---|---|---|---|---|---|---|---|---|---|
| F-score↑ | 0.09 | **0.17** | 0.32 | **0.34** | 0.46 | **0.47** | 0.40 | **0.42** | **0.52** | 0.52 |
| Planar Vertices | 317.5K | **4.53K** | 609.3K | **6.94K** | 3.04M | **41.26K** | 503.1K | **6.89K** | 2.39M | **29.27K** |
| Overall Mesh Vertices | 1.86M | **1.55M** | 3.75M | **3.03M** | 57.82M | **53.58M** | 2.39M | **1.76M** | 14.69M | **12.04M** |

Table 2: Quantitative evaluations on Tanks and Temples Dataset Knapitsch et al. (2017).

**Implementation Details** We implement our GSPlane method on five representative GS-based methods, including 3DGS Kerbl et al. (2023), 2DGS Huang et al. (2024a), GOF Yu et al. (2024), RaDe-GS Zhang et al. (2024), and PGSR Chen et al. (2024). The initial mesh is extracted with the proposed process from the baseline, with the voxel size as 0.005. Note that GOF adopt the Marching Tetrahedral meshing strategy as their baseline, in comparison we have conducted experiments on GOF with both Marching Tetrahedral and TSDF to extract reconstructed meshes, and report both of the evaluation metrics in the Tables. During the experiment, we set the threshold of cosine similarity $\alpha$ to 0.98.

## 4.2 OVERALL PERFORMANCE

The indoor quantitative results of the overall metrics are presented in Tab. 1. Specifically, *Ours-train* denotes applying structured representation of planes and Dynamic Gaussain Re-classifier in the training stage, while *Ours-full* further incorporates mesh layout refinement in the post-training stage. Note that the **Supportive Plane Correction** (SPC) step is excluded from the performance evaluation. For a fair comparison, we also report results from GaussianRoom Xiang et al. (2024) and AlphaTablets He et al. (2024), which leverage normal maps, depth, and edge information as priors for reconstruction. Compared with the methods that adopt off-the-shelf predictions for direct supervision, our GSPlane demonstrates the effectiveness of incorporating planar priors. The results highlight that the structured plane representation consistently improves both geometric and rendering quality across baselines, while the proposed mesh layout refinement enables more accurate and complete surface estimation. *Ours-train* achieves a slight reduction in vertex count compared to baseline methods because it produces tighter and more compact planar distribution of Gaussians, while *Ours-full* significantly reduces the number of vertices in the final mesh. Notably, the structured Gaussian planar representation also contributes to enhanced rendering quality, see Sec. D in Appendix for rendering visualizations.

The outdoor quantitative results are displayed in Tab. 2, where we report the F-score as the reconstruction metric, along with the number of planar and total vertices for comparison. *Ours* in Tab. 2 corresponds to the *Ours-full* configuration in Tab. 1. As seen in the table, our method improves reconstruction performance in outdoor scenes while significantly reducing the number of vertices in the mesh. However, the geometric improvements are less pronounced compared to indoor scenes, primarily because the TNT dataset contains fewer planar regions in some scenarios compared to

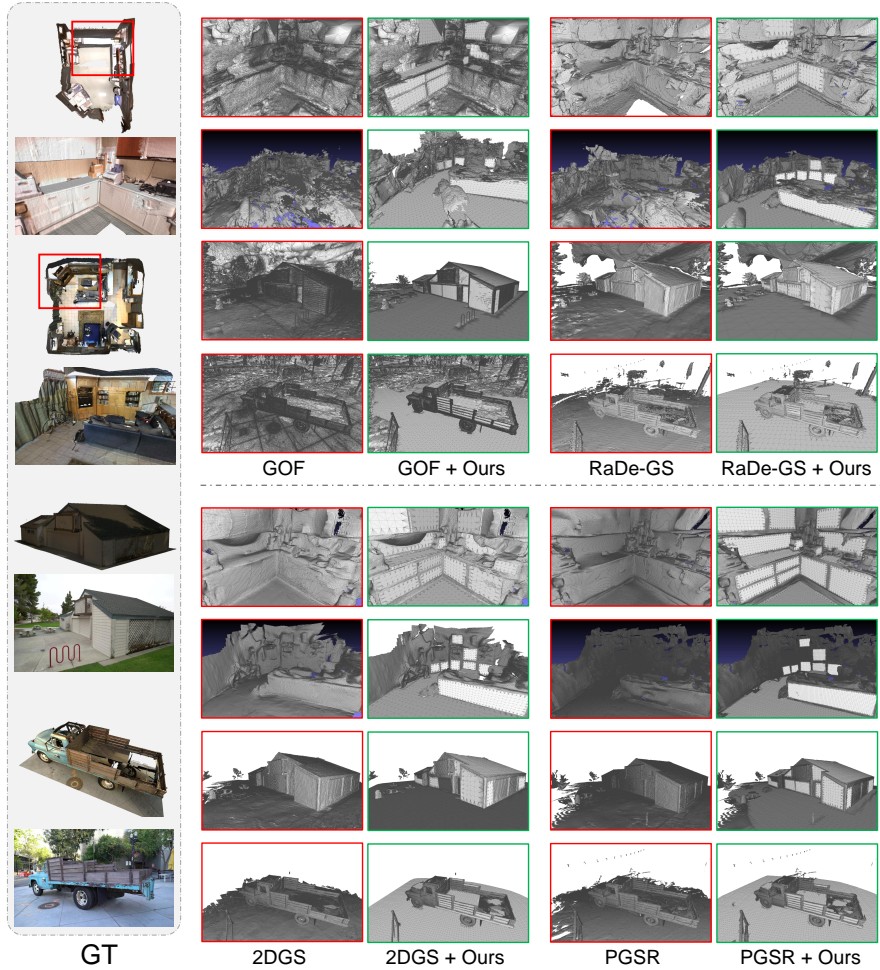

GOF      GOF + Ours      RaDe-GS      RaDe-GS + Ours

GT      2DGS      2DGS + Ours      PGSR      PGSR + Ours

Figure 3: Visualizations of the mesh performance on both indoor and outdoor scenes. We provide comparisons on four baseline methods. We provide default geometries and wireframes of the reconstructed meshes from baseline results on the left, and our refined wireframes as comparisons on the right. It can be seen that our method can reduce the number of vertices by large margin, while maintaining consistent normal and topology across different planes. More examples can be found in Appendix.

ScanNetV2. Moreover, the TNT dataset's ground truth lacks explicit connectivity and planar-related annotations, making it challenging to fully evaluate the strengths of our method, which is specifically designed to deliver accurate and smooth topology on planar surfaces. Nevertheless, our method still achieves substantial reductions in mesh vertex count, demonstrating its efficiency in outdoor settings. Visualizations for both indoor and outdoor scenes can be found in Fig. 3.

## 4.3 PLANAR-WISE GEOMETRY

The planar metrics, including Fidelity, Accuracy, and L1-Chamfer Distance, are presented in Tab.3. Our proposed planar-guided mesh extraction demonstrates significant potential for improving the reconstruction of planar regions across various Gaussian Splatting baselines. We also provide visualizations starting from 2D planar prior to the final refinement results in Fig.4. Before training, we first establish planar priors by aggregating both subparts proposals from SAM Kirillov et al. (2023) and normal maps from Metric3Dv2 Hu et al. (2024). After structured representation for 3D planes are established, given a unrefined mesh with densely distributed vertices, GSPlane can create refined planar regions that exhibit consistent normals and topology, along with unified edges and a reduced number of vertices and faces, resulting in a more efficient and structured representation.

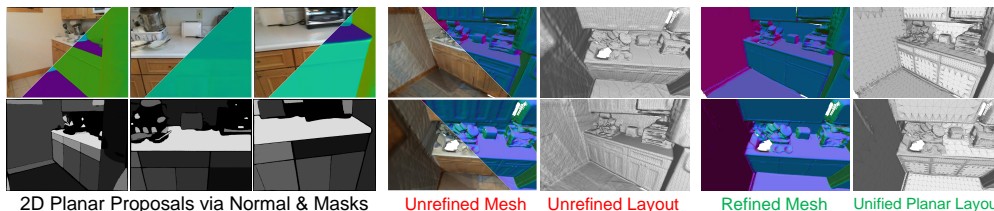

2D Planar Proposals via Normal & Masks | Unrefined Mesh  Unrefined Layout | Refined Mesh  Unified Planar Layout

Figure 4: Visualization of an example from kitchen corner. The left shows the normal map and aggregated planar mask proposals of 2D views. The middle and right of the figure are the target mesh before & after the layout refinement via structured representation of planes.

## 4.4 ABLATION STUDY

We conduct an ablation study to evaluate the effectiveness of different modules in GSPlane, including the optimization of basis points, the **Dynamic Gaussian Re-classifier** (DGR), and the post-refinement of the mesh layout. The results are presented in the left table of Fig. 5. Compared to the baseline performance of 2DGS, our GSPlane significantly enhances the quality of the generated mesh. It is worth pointing out that the proposed DGR module only shows minor improvements on the final metrics. This phenomenon meets our expectation, as it reveals that the planar priors extracted from off-the-shelf models (*i.e.* SAM, Metric3DV2) are reasonably reliable. However, DGR still remains a crucial component to handle those rare but impactful errors, ensuring robustness in scenarios where misclassifications might otherwise degrade performance.

Additionally, we perform experiments on 3 different baselines, including 2DGS, RaDe-GS, and PGSR, all of which estimate normal maps during the rasterization process. Our goal is to analyze the differences between our proposed structured representation and directly using off-the-shelf normal maps to supervise the estimated normals. As shown in the right table of Fig. 5, our structured planar representation outperforms results with direct monocular normal supervision in 2DGS and PGSR, while in RaDe-GS it shows comparable. This is because that the workflow in RaDe-GS are less stable when encountering low resolution samples from ScanNet, resulting in poor geometry (compared with 2DGS and PGSR in Fig. 3). Our structured representation for planes surpasses the direct supervision of monocular normal in most of the baselines.

We also provide detailed ablations on the hyperparameter of the DGR module. We choose two different baselines, 2DGS and PGSR, to run on ScanNet samples for ablations. The training setting follows the same setting in the left table of Fig. 5 '+ Train', which exclude our mesh layout refinement. The results are shown in Fig. 6. Overall, 5% and 20% yields promising results among different settings, and show its robustness across different baselines.

For ablation studies on more hyperparameters, please refer Tab. 4 and Tab. 5 in Appendix Sec. A.

| Method | Fidelity↓ | Acc↓ | CD↓ |
|---|---|---|---|
| PlanarRecon Xie et al. (2022) | 18.86 | 16.21 | 17.53 |
| AirPlanes Watson et al. (2024) | 8.76 | 7.98 | 8.37 |
| PlanarSplatting Tan et al. (2025) | 6.64 | 11.76 | 9.2 |
| 3DGS + Ours-full | 6.21 / 6.75 | 7.95 / 8.15 | 7.08 / 7.35 |
| 2DGS + Ours-full | **5.49** / 5.82 | 7.32 / 7.71 | 6.41 / 6.77 |
| GOF + Ours-full | 8.25 / 8.74 | 9.50 / 9.93 | 8.88 / 9.34 |
| RaDe-GS + Ours-full | 7.57 / 7.83 | **6.34 / 6.60** | 6.96 / 7.22 |
| PGSR + Ours-full | 5.24 / **5.39** | 6.58 / 6.65 | **5.91 / 6.02** |

Table 3: Planar-wise metrics evaluated on $k = 20/k = 30$ largest plane regions from gt mesh in ScanNetV2, following Airplanes Watson et al. (2024). The results from methods displayed in grey are evaluated with $k = 20$ from the papers.

## 4.5 APPLICATION ON SUPPORTIVE PLANE

To validate the effectiveness of **Supportive Plane Correction** (SPC), we conducted experiments demonstrating its ability to accurately reconstruct supportive planes and decouple objects resting on them. As shown in the left of Fig. 7, the default result of mesh layout refinement can provide unified grid points on plane, but the boundaries of the placed object are connected with the grid points to

| | Prec↑ | Recall↑ | F-score↑ |
| --- | --- | --- | --- |
| 2DGS | 0.603 | 0.568 | 0.583 |
| + Train w/o basis points | 0.637 | 0.596 | 0.616 |
| + Train w/o DGR | 0.648 | 0.613 | 0.630 |
| + Train | 0.650 | 0.620 | 0.633 |
| + Train + Mesh Ref. | 0.664 | 0.716 | 0.689 |

| Setting | Acc↓ | Comp↓ | Prec↑ | Recall↑ | F-score↑ |
| --- | --- | --- | --- | --- | --- |
| 2DGS Huang et al. (2024a) | 0.0661 | 0.0782 | 0.6035 | 0.5676 | 0.5834 |
| 2DGS + normal | 0.0645 | 0.0764 | 0.6396 | 0.5972 | 0.6177 |
| 2DGS + Ours-train | **0.0630** | **0.0733** | **0.6501** | **0.6197** | **0.6330** |
| RaDe-GS Zhang et al. (2024) | 0.1008 | 0.1041 | 0.4805 | 0.5069 | 0.4914 |
| RaDe-GS + normal | **0.0947** | 0.1024 | **0.5179** | 0.5388 | 0.5281 |
| RaDe-GS + Ours-train | 0.0960 | **0.1016** | 0.5069 | **0.5576** | **0.5283** |
| PGSR Chen et al. (2024) | 0.0791 | 0.0852 | 0.5815 | 0.5711 | 0.5733 |
| PGSR + normal | 0.0732 | 0.0710 | 0.6124 | 0.6095 | 0.6109 |
| PGSR + Ours-train | **0.0654** | **0.0633** | **0.6332** | **0.6401** | **0.6345** |

Figure 5: Ablation study results on GSPlane. The left table shows the effectiveness of different modules in GSPlane, and the right table compares our structured representation with off-the-shelf normal map supervision for mesh geometry reconstruction.

| Top % planar (with non-planar @ 20%) | Prec↑ | Recall↑ | F-score↑ | Final Ratio (num of planar / all gaussians) |
| --- | --- | --- | --- | --- |
| 1% | 0.649 | 0.618 | 0.632 | 0.19 |
| 2% | **0.650** | 0.616 | 0.632 | 0.19 |
| 5% | **0.650** | **0.620** | **0.633** | 0.17 |
| 10% | 0.648 | 0.615 | 0.631 | 0.13 |

(a) 2DGS

| Top % non-planar (with planar @ 5%) | Prec↑ | Recall↑ | F-score↑ | Final Ratio (num of planar / all gaussians) |
| --- | --- | --- | --- | --- |
| 5% | 0.649 | 0.617 | 0.633 | 0.18 |
| 10% | 0.651 | **0.620** | 0.633 | 0.18 |
| 15% | **0.653** | 0.618 | **0.635** | 0.18 |
| 20% | 0.650 | **0.620** | 0.633 | 0.17 |
| 30% | 0.647 | 0.616 | 0.632 | 0.15 |

(b) 2DGS

| Top % planar (with non-planar @ 20%) | Prec↑ | Recall↑ | F-score↑ | Final Ratio (num of planar / all gaussians) |
| --- | --- | --- | --- | --- |
| 1% | 0.631 | 0.637 | 0.632 | 0.19 |
| 2% | **0.633** | 0.638 | 0.633 | 0.18 |
| 5% | **0.633** | **0.640** | **0.634** | 0.18 |
| 10% | 0.632 | 0.637 | 0.633 | 0.15 |

(c) PGSR

| Top % non-planar (with planar @ 5%) | Prec↑ | Recall↑ | F-score↑ | Final Ratio (num of planar / all gaussians) |
| --- | --- | --- | --- | --- |
| 5% | 0.629 | 0.635 | 0.630 | 0.21 |
| 10% | 0.631 | 0.637 | 0.632 | 0.19 |
| 15% | 0.631 | 0.639 | 0.633 | 0.18 |
| 20% | **0.633** | **0.640** | **0.634** | 0.18 |
| 30% | 0.628 | 0.631 | 0.630 | 0.14 |

(d) PGSR

Figure 6: Ablation studies on the DGR hyperparameter. Table (a) and (b) are conducted on 2DGS baseline, (c) and (d) are conducted on PGSR baseline. From the tables we can see that 5% and 20% yields promising results among different settings, and show its robustness across different baselines.

maintain wholeness of the structure. By fully utilizing the optimized planar priors, it is possible to infer the real shape and structure of the supportive plane - desk, and objects placed on the desk can also be removed from the desk. This ensures that the reconstructed supportive plane remains continuous and free of artifacts, even in the presence of complex void geometries. The hole of the objects at the contact area can also be sealed using the supportive plane function, and are further free to manipulate across the supportive plane or within the scene.

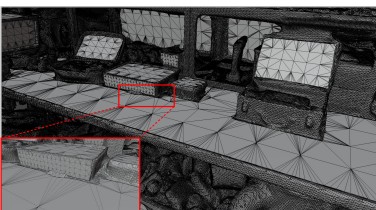 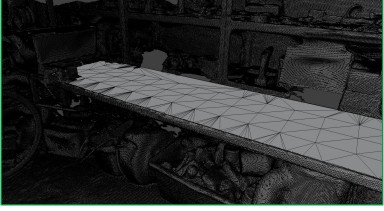 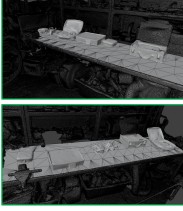

Mesh Layout Refinement (w/o SPC)     Reconstruction of Desk (Supportive)     Decoupled Objects

Figure 7: Visualizations of Supportive Plane Correction. When running SPC, the object boundaries are excluded from plane reconstruction, leading to an intact plane with complete shape like in reality. The objects are decoupled from the supportive plane surface, and can be further moved or manipulated freely.

## 5 CONCLUSION

In this paper, we highlight the potential of incorporating plane prior knowledge into Gaussian Splatting for improved reconstruction of planar regions. By leveraging segmentation and surface normal estimation, GSPlane generates structured planar representations, improving the geometric accuracy and topological consistency of meshes while reducing the density of vertices and faces. Additional discussion on supportive plane demonstrates that our structured planar representation enables realistic plane completion and decouples objects from planes, allowing further object manipulation. Our experiments demonstrate that leveraging this prior significantly enhances the geometric accuracy and topological consistency of extracted meshes, reducing the complexity of the mesh structure.

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

## A  ADDITIONAL ILLUSTRATION ON STRUCTURED REPRESENTATION FOR PLANES

This section provide additional illustrations on how the structured representation for planes are obtained from 2D images. After obtaining the normal maps and subpart mask proposals from off-the-shelf models, we first multiply the normal map $N_i$ with each mask proposal $M_{i,j}$, where $i$ denotes the $i$-th image and $j$-th mask, to isolate the normal distribution $N_{mask}$ within each instance region. To determine if a region is planar, we take cosine similarity to measure the distance between each pixel normal to the average normal within the instance. Empirically, if more than 70% of the pixels have a similarity larger than a certain threshold $\alpha$, we regard these pixels as a single plane. The largest connected region of these valid pixels is then selected as a plane proposal. In case multiple planes are mistakenly segmented into a single mask and do not meet the previous condition, we apply K-means clustering to the normals in this region with pixel number bigger than $\sigma$. We then evaluate each cluster using the 70% criterion to identify all potential planes. If none of the clusters meet the criterion, the mask proposal is considered non-planar. In our experience, setting the target number of clusters to 2 yields good results. By following these steps, we can identify all the plane proposals $M^i_{plane}$ in image $I_i$.

To address the potential intersections among the obtained plane proposals $\mathcal{M}^i_{plane}$, we implement a series of steps to resolve conflicts in these overlapping areas. We first define an empty list $\mathcal{M}^i_{merge}$ to store the exclusive planar masks after the process. We iteratively select each element $M^{i,k}_{plane}$ in $\mathcal{M}^i_{plane}$, and compute normal vector cosine similarity with all other proposals $M^{i,l}_{plane}$. If any proposals matches through aforementioned 70% criteria, they are merged together with $M^{i,k}_{plane}$ and pop out from $\mathcal{M}^i_{plane}$. The final $M^{i,k}_{plane}$ will be stored in $\mathcal{M}^i_{merge}$. After completing all the planar proposals in $\mathcal{M}^i_{plane}$, we achieve a collection of mutually exclusive planar masks $\mathcal{M}^i_{merge}$. By assigning each element with an index, we are able to obtain the final planar mask $P_i$. The overall algorithm is detailed in Alg. 1.

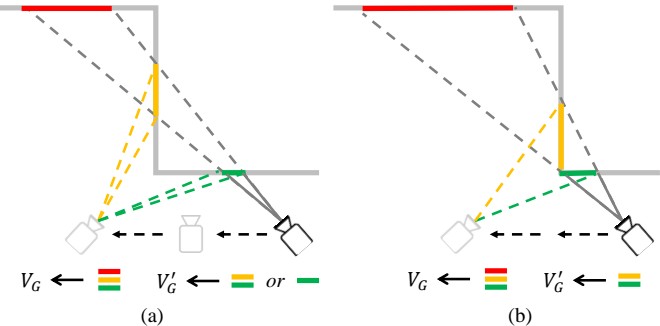

Figure 8: Illustration of 2 possible situations when encountering occlusion. Here red region and yellow region are denoted as occluded points as they are not visible in the camera. In both situation, the red region will be filtered by clustering the depth information.

When lifting 2D priors into 3D space, given a planar instance map $P_i$ with corresponding extrinsics $[R_i, t_i]$, and the intrinsic matrix $K$, we begin by projecting all nodes $V$ back into 2D camera coordinates. For each plane instance indicated in $P_i$, there is a group of points $V_G$ projected onto this region. We perform K-means clustering on projected depths with $K = 2$ to coarsely filter out occluded points that may not appear in the image. An illustration figure of this process is shown in Fig. 8. The occluded points will be projected onto the plane region together with the foreground points. We only consider the closest as plane-related points in each camera pose, so filtering out points with larger depth is necessary. Points with similar depths in one camera can be further distinguished through other views. The filtered point set is denoted as $V'_G$, and the edge $E(V_x, V_y \in V'_G)$ will be established among these points, as they are considered to be in the same plane from the plane instance $P_i$. For every two nodes $V_x, V_y \in V'_G$, the edge $E(V_x, V_y)$ will be created with the weight of 1 if it doesn't exist before. Otherwise, the weight will be incremented by 1. Using Leiden Algorithm to divide different communities, we identify the Gaussians distributed across each plane in the scene.

**Algorithm 1** 2D Planar Perception

---

**Require:** normal map $N_i$, mask proposals $\{M_{i,j}\}$

1: **for** each $M_{i,j}$ **do**

2: $\quad N_{\text{mask}} \leftarrow N_i \odot M_{i,j},$

3: $\quad d \leftarrow cos\_sim(N_{\text{mask}}, \overline{N_{\text{mask}}})$

4: $\quad$ **if** $\text{ratio}(d > \alpha) < 0.3$ **then**

5: $\quad\quad M_{plane}^{i,j} \leftarrow M_{i,j}[d > \alpha]$

6: $\quad$ **else if** $\text{Area}(N_{mask}) > \sigma$ **then**

7: $\quad\quad N_{cluster1}, N_{cluster2} \leftarrow \text{K-means}(N_{mask})$

8: $\quad\quad$ Repeat Step 2-5 on $N_{cluster1}, N_{cluster2}$

9: $\quad$ **end if**

10: **end for**

11: $\mathcal{M}_{plane}^i \leftarrow [M_{plane}^{i,j}]$

12: $\mathcal{M}_{merge}^i \leftarrow$ empty list

13: **while** $\mathcal{M}_{plane}^i$ not empty **do**

14: $\quad M_{\text{plane}}^{i,k} \leftarrow \mathcal{M}_{plane}^i[0]$

15: $\quad$ **for** each $l \neq k$ **do**

16: $\quad\quad d' \leftarrow cos\_sim(\overline{M_{\text{plane}}^{i,k}}, \overline{M_{plane}^{i,l}})$

17: $\quad\quad$ **if** $d' > \alpha$ **then**

18: $\quad\quad\quad M_\cap \leftarrow M_{\text{plane}}^{i,k} \cap M_{plane}^{i,l}$

19: $\quad\quad\quad M_{\text{plane}}^{i,k} \leftarrow M_{\text{plane}}^{i,k} + M_{plane}^{i,l} - M_\cap$

20: $\quad\quad\quad \mathcal{M}_{plane}^i.\text{pop}(M_{\text{plane}}^{i,l})$

21: $\quad\quad$ **end if**

22: $\quad$ **end for**

23: $\quad \mathcal{M}_{merge}^i.push(M_{\text{plane}}^{i,k})$

24: **end while**

25: $P_i \leftarrow$ assign instance ID with $\mathcal{M}_{merge}^i$

26: **return** $P_i$

---

The ablation studies for hyperparameter $\alpha, \sigma$ are displayed in Tab. 4 and Tab. 5. Here, we choose RaDe-GS as the baseline method, and run full settings of GSPlane. When implementing our experiments, we choose $\alpha = 0.98$ and $\sigma = 200$ as our settings.

| $\alpha$ | Acc↓ | Comp↓ | Prec↑ | Recall↑ | F-score↑ | *num_plane* |
|---|---|---|---|---|---|---|
| 0.95 | **0.0821** | 0.0861 | 0.5168 | 0.672 | 0.5842 | 35.57 |
| 0.98 | 0.0824 | 0.0855 | 0.5197 | **0.6738** | **0.5868** | 34.43 |
| 0.99 | 0.0831 | **0.0829** | **0.5214** | 0.6654 | 0.5846 | 31.29 |

Table 4: Ablation on the cosine similarity threshold $\alpha$.

| $\sigma$ | Acc↓ | Comp↓ | Prec↑ | Recall↑ | F-score↑ | *num_plane* |
|---|---|---|---|---|---|---|
| 100 | **0.0824** | **0.0855** | **0.5197** | **0.6738** | **0.5868** | 34.43 |
| 200 | **0.0824** | **0.0855** | **0.5197** | **0.6738** | **0.5868** | 34.43 |
| 500 | 0.0827 | 0.0874 | 0.5175 | 0.6699 | 0.5839 | 32.14 |

Table 5: Ablation on the minimum pixel number $\sigma$ of K-means clustering.

## B  ALGORITHMIC ILLUSTRATION ON MESH LAYOUT REFINEMENT

---

**Algorithm 2** Mesh Layout Refinement

---

**Require:** Extracted mesh $O$, Initial sparse point cloud $Pcd$, voxel size $\delta$, precomputed planar relationships $V_P^i \in Pcd$
1: **for** each plane $A \in V_P$ **do**
2:     **for** each vertex $v_x \in O$ **do**
3:         **if** $\exists v_y \in V_P^A, |v_y - v_x| < 1.5\delta$ **and** $\forall \bar{v}_y \notin V_P^A, |\bar{v}_y - v_x| > 0.5\delta$ **then**
4:             Assign $v_x$ to plane $A$ in $O$: $v_x \to \hat{V}_P^A \in O$
5:         **end if**
6:     **end for**
7: **end for**
8: **for** each $\hat{V}_P^A$ **do**
9:     Remove planar faces: $\{f \in O | f = (v_1, v_2, v_3), v_1, v_2, v_3 \in \hat{V}_P^A\}$
10:     Categorize vertices: Boundary $\hat{V}_B^A$, Interior $\hat{V}_I^A$
11:     Project $\hat{V}_B^A, \hat{V}_I^A$ onto plane $A$: $\hat{V}_B^A \to \hat{V}_B^a, \hat{V}_I^A \to \hat{V}_I^a$
12:     Compute bounding rectangle $R_A$ covering $(\hat{V}_B^a, \hat{V}_I^a)$ and generate grid points $G_A$ within $R_A$
13:     Exclude $G_A$ points outside the projected region $(\hat{V}_B^a, \hat{V}_I^a)$
14:     Perform Delaunay triangulation: $T_A = \text{Delaunay}(V_B^A \cup G_A)$
15:     Map $T_A$ and $G_A$ back to 3D space
16:     Integrate $T_A, G_A$ into $O$
17: **end for**
18: **return** Refined mesh $O'$

---

## C  IMPLEMENTATION OF DYNAMIC GAUSSAIN RE-CLASSIFIER

This section we provide some implementation details of our Dynamic Gaussian Re-classifier (DGR). The DGR is designed to identify and reclassify Gaussians that are mistakenly regarded as planar Gaussians. According to the general design of Gaussian training process, the distribution of Gaussians will be densified from Iteration 500 to 15,000 in each 100 iteration, and the whole training process will end at Iteration 30,000. Our DGR phase will be operating for the latter 50 iterations between every densification step, and for 100 iterations at Iteration 20,000.

During the DGR phase, gradients of both planar Gaussians and non-planar Gaussians before finally proceeding to back-propagation will be stored and averaged for evaluation. The top 5% of the planar gradients are selected and compared with the average magnitude of top 20% non-planar gradients.

Those with higher gradient magnitudes, the coordinates of their corresponding planar Gaussians will be re-formulated back to $xyz$ format. The DGR design can correct those mistaken planar Gaussians, and it will not influence the training for non-planar Gaussians. Thus, even if the true-positive planar Gaussians are processed, they will still be supervised with the baseline design.

# D  ADDITIONAL QUALITATIVE RESULTS ON NOVEL VIEW SYNTHESIS

In this section we provide examples in both rendering effects of GSPlane and baseline methods in Novel View Synthesis. The visualizations are shown in Fig. 9. According to the quantitative results in Tab. 1, GSPlane also provides comparable results with small improvements, up to 0.018 and 1.02 for GOF in the SSIM and PSNR, respectively.

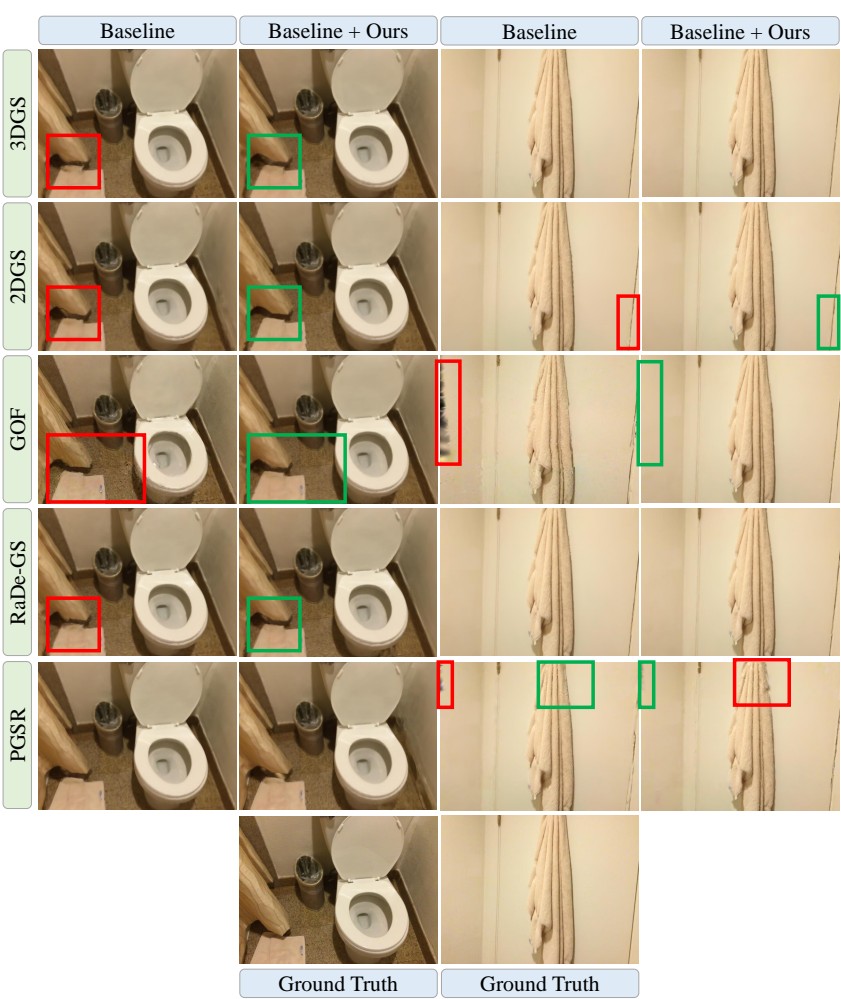

Figure 9: Visualization of NVS results.

## E  EFFICIENCY OF GSPLANE

In order to fully evaluate the reliability of GSPlane, we further conduct experiments to measure the time overhead for each step of our workflow. Results are presented in Fig 6, conducted on a single A100-40GB. From the table, the first time of extracting planar prior may take several hours to proceed, but the time overhead of our structured planar representation will only bring around 8 minutes in training, and around half a minute for mesh extraction.

| Planar Prior | SAM(*per img*) | Metric3D(*per img*) | 2D prior (*per img*) | 3D prior |
|---|---|---|---|---|
| Processing | 14.72s | 4.96s | 2.78s | 182.45s |
| Training | Initialization | Densify | Training | Save Ply |
| | +5.91s | +2.80s | +473.68s | +2.04s |
| Mesh | 2DGS | GOF(Tetra) | PGSR | GSPlane |
| Extraction | 43.52s | $8 \times 494.17$s | 71.49s | + 37.71s |

Table 6: Time overhead for the proposed **GSPlane**.

## F  VISUAL COMPARISON WITH VGGT

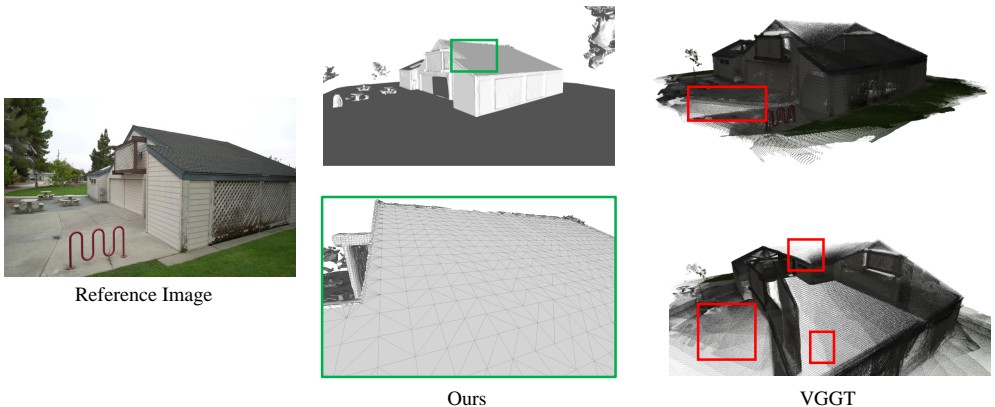

Reference Image

Ours

VGGT

Figure 10: Visual comparison between our GSPlane and VGGT reconstruction on scene 'Barn'.

We further conduct experimental studies to compare the reconstruction performance of our GSPlane with VGGT Wang et al. (2025). The visualizations are shown in Fig. 10. VGGT is a powerful benchmark that empowers multiple 3D tasks, but its workflow on generating dense reconstruction is still less satisfactory. First, VGGT only accept limited number of images as input, where only 1/4 of the image source could be implemented when reconstructing 'Barn'. Second, the generated reconstruction only consists of discrete points without any connectivity or topology among them. Third, the point map is located in an extremely uneven density with unexpected visual edges, which is caused by camera angle distributions. Compared with VGGT, our GSPlane can obtain unified, well-organized geometry and topology on planar surfaces with the help of planar priors.

## G  LIMITATION

Though GSPlane is able to provide concise and accurate geometry with satisfied topology and unified normal in planar region, there are still some issues before acquiring a desired and satisfied scene mesh. Currently, our focus is on planar regions, and the structured representation of non-planar regions remains an open challenge, which we leave as future work. A possible direction for addressing this issue could involve developing alternative representations tailored to complex surfaces. Additionally, the accuracy of planar priors are constrained by foundation models of masks and normals.

## H   LARGE LANGUAGE MODEL USAGE

Large Language Models (LLMs) are used for polishing writing in this manuscript. The prompt is used as follows:

*Assume you are a native English speaker, a senior researcher in the area of computer vision and graphics. Please help me polish the following content:* ___

