# OpenReview forum: "GSPlane: Concise and Accurate Planar Reconstruction via Structured Representation"
_ICLR.cc/2026/Conference — Submitted to ICLR 2026_

### Official Review · Reviewer_VWBM · 2025-10-30

**Soundness:** 3
**Presentation:** 2
**Contribution:** 3
**Rating:** 6
**Confidence:** 3

**Summary:**

GSPlane injects 2D planar priors (SAM masks + Metric3Dv2 normals) into Gaussian Splatting by re-parameterizing planar Gaussians as convex combinations of three learned basis points; a Dynamic Gaussian Re-classifier (DGR) reverts false planar assignments; mesh layout refinement and an optional Supportive Plane Correction (SPC) yield cleaner, topology-consistent planar meshes while preserving NVS quality on ScanNetV2 and Tanks & Temples with large vertex reductions.

**Strengths:**

1. Elegant plane parameterization (weights over three basis points) that directly enforces coplanarity during training.
2. DGR automatically corrects misclassified planar Gaussians without hurting non-planar training.
3. Clear practical gains: big drops in planar/overall vertex counts with comparable or slightly better NVS metrics.

**Weaknesses:**

1. Dependence on off-the-shelf priors. Quality hinges on SAM/Metric3Dv2; the authors acknowledge the limitation and domain-shift sensitivity.
2. Heuristic thresholds in DGR. The 5%/20% gradient rule is plausible but somewhat ad hoc; stability vs. missed corrections is not deeply analyzed.
3. Comparability caveat for GOF. For GOF they replace Marching Tetrahedra with TSDF fusion “to avoid mesh in actually empty areas,” which may confound apples-to-apples comparisons.

**Questions:**

1. Basis-point stability: How do you prevent degeneracy/collinearity of the three basis points per plane during training, and do you re-seed them if the plane rotates significantly?
2. Fairness of meshing choices: Can you report GOF results with both (a) original meshing and (b) TSDF, to isolate GSPlane’s contribution from mesher effects?

---

> ### Author Response · Authors · 2025-11-21
>
> We are sincerely grateful for your understanding and attentive review on our manuscript, and your concerns and questions are well received. The responses of the listing weaknesses and questions are as follows:
>
> # 1.	DGR and Prior dependence
>
> When extracting planar priors before training, there will be two types of mis-classifications: Non-planar points defined as planar, and planar points defined as non-planar. Consider that the non-planar gaussians will be supervised under baseline methods (i.e. default 2DGS, GOF, RaDe-GS), and thus will not cause heavily deviation, those that are mistakenly bound on the plane will certainly cause more trouble. Hence, we propose DGR to remove those points. According to the left table in Fig.5 from the manuscript, adding DGR only brings slight improvements on the overall score. Here we provide more ablations on discussing how different percentage will affect the results. Statistics also include the final ratio of planar and non-planar gaussians, so as to indicate the amount of removal happening in DGR.
> | Top % planar (with non-planar @ 20%) | Prec ↑ | Recall ↑ | F-score ↑ | Final Planar Ratio |
> |--------------------------------------|--------|----------|-----------|--------------------------------------------|
> | 1%                                   | 0.649  | 0.618    | 0.632     | 0.19                                       |
> | 2%                                   | **0.650**  | 0.616    | 0.632     | 0.19                                       |
> | 5%                                   | **0.650**  | **0.620**    | **0.633**     | 0.17                                       |
> | 10%                                  | 0.648  | 0.615    | 0.631     | 0.13                                       |
>
> | Top % non-planar (with planar @ 5%) | Prec ↑ | Recall ↑ | F-score ↑ | Final Planar Ratio |
> |--------------------------------------|--------|----------|-----------|--------------------------------------------|
> | 5%                                   | 0.649  | 0.617    | 0.633     | 0.18                                       |
> | 10%                                   | 0.651  | **0.620**    | 0.633     | 0.18                                       |
> | 15%                                   | **0.653**  | 0.618    | **0.635**     | 0.18                                       |
> | 20%                                  | 0.650  | **0.620**    | 0.633     | 0.17                                       |
> | 30%                                  | 0.647  | 0.616    | 0.632     | 0.15                                       |
> *Above are results conducted on 2DGS baseline with ScanNet samples.*
>
> | Top % planar (with non-planar @ 20%) | Prec ↑ | Recall ↑ | F-score ↑ | Final Planar Ratio |
> |--------------------------------------|--------|----------|-----------|--------------------------------------------|
> | 1%                                   | 0.631  | 0.637    | 0.632     | 0.19                                       |
> | 2%                                   | **0.633**  | 0.638    | 0.633     | 0.18                                       |
> | 5%                                   | **0.633**  | **0.640**    | **0.634**     | 0.18                                       |
> | 10%                                  | 0.632  | 0.637    | 0.633     | 0.15                                       |
>
> | Top % non-planar (with planar @ 5%) | Prec ↑ | Recall ↑ | F-score ↑ | Final Planar Ratio |
> |-------------------------------|--------|----------|-----------|--------------------------------------------|
> | 5%                                   | 0.629  | 0.635    | 0.630     | 0.21                                       |
> | 10%                                   | 0.631  | 0.637    | 0.632     | 0.19                                       |
> | 15%                                   | 0.631  | 0.639    | 0.633    | 0.18                                       |
> | 20%                                  | **0.633**  | **0.640**    | **0.634**     | 0.18                                       |
> | 30%                                  | 0.628  | 0.631   | 0.630     | 0.14                                       |
> *Above are results conducted on PGSR baseline with ScanNet samples.*
>
> The results listed in the table should be compared with the setting of ‘+ Train’ in the left table of Fig.5. From the tables it can tell that 5% and 20% yields promising results among different settings, and show its robustness across different baselines. Results are included in the main body of the updated manuscript in Fig.6.
> When considering the off-the-shelf model performances, while these models may fail to detect all planes due to their limited performance, which could potentially hinder the upper bound of our method, our approach is capable of avoiding the errors introduced by these models, with the help of DGR. The limited improvement DGR brought to the performance could also be served a support for this point.

---

> ### Author Response · Authors · 2025-11-21
>
> # 2.	GOF issue
> We understand that the unfair comparisons have caused some controversies. We adopt the evaluation metrics from GOF that output meshes will be cropped to exclude areas away from GT, and update the GOF results in Tab.1 including both Marching Tetrahedra and TSDF fusion statistics. Note that for GOF visualizations in Fig.3, the “GOF + Ours” is processed on the Marching Tetrahedra output and already cropped faraway areas, so the visualizations can remain unchanged. Due to one more page vacancy now, more visualizations are provided, please refer to the updated PDF file if interested.
>
> # 3.	Basis-point stability
> When sampling basis points at first, there will be restriction that the distance between each point should exceed 1/4 planar area diameter. During training, for each iteration, there will be verification to ensure basis points are not collinear, otherwise the basis points will be resampled following the same restriction. As far as our existing experiments, there is no significant rotation occurs during training, partly because we keep a distance between basis points to prevent close points with large oscillation. Details have been added to the updated manuscript in Sec. 3.1.

---

> ### Comment · Reviewer_VWBM · 2025-11-27
>
> Thank you to the authors for addressing my concerns; I maintain my original score.

---

> > ### Author Response · Authors · 2025-11-27
> >
> > We're glad our response addressed your questions. Thank you again for reviewing our work and providing valuable suggestions!

---

### Official Review · Reviewer_gBHm · 2025-10-31

**Soundness:** 2
**Presentation:** 3
**Contribution:** 2
**Rating:** 4
**Confidence:** 4

**Summary:**

This work focuses on improving surface reconstruction of Gaussian Splatting (GS) with structured plane representations. To this end, the method extracts plane priors based on monocular normal predictions and SAM model. Based on plane priors, the method defines structured plane representations by using a normalized linear combination of three non-collinear points. With this representation, the method introduces a dynamic Gaussian re-classifier to correct false-positive Gaussians based on the average gradients of planar Gaussians and non-planar Gaussians. In addition, the method mesh simplification based on the structured representation. The experiments show that the method can improve surface reconstruction of different GS methods for indoor scenes and outdoor scenes.

**Strengths:**

1. The method defines a structured plane representation to constrain the Gaussian points on the same plane, different from the squeezing Gaussians as planes in 2DGS and PGSR.
2. The introduced plane representation can help simplify meshes.
3. The method can improve different GS surface reconstruction on ScanNet dataset.

**Weaknesses:**

1. Althought the introduced plane representation can help simplify meshes, it needs to extract meshes by TSDF fusion first. This means that it cannot direclty extract lightweight meshes, and  aslo require large storage.
2. According to the experiments in Tables 1 and 2, the introduced mesh refinement improves surface reconstrucion a lot. However, the overall improments on Tanks and Temples (Table 2) are limited. Therefore, I wonder maybe some components do not work in Tanks and Temples dataset.
3. According to the right table in Figure 4, if SOTA GS methods direclty use monocular normal estimation as supervisions, their performance is very competitive with the proposed method. In this way, the advantages of the introduced structured plane representation seem somewhat weak.

**Questions:**

1. For Tanks and Temples dataset, the improvements of the method are very limited. It is better to show the performance without mesh refinement. This can help understand what components are most important in the method.
2. For the right table of Figure 4, if PGSR is combined with monocular normal supervision, how about the performance? I suspect the performance improvement of the method comes from the constraint of the monocular normal estiamtion.
3. It is better to compare the effieciency of different methods.
4. For the dynamic Gaussian re-classifier, accodring to the left table in Figure 4, its boost is very limited. I wonder if it works for other GS baselines. Moreover, in Line 235-236, how to determine the thresholds, 5% and 20%?

---

> ### Author Response · Authors · 2025-11-21
>
> We appreciate your effort in reviewing our manuscript, and the comments that can help us improve our work! The response for the weaknesses and questions are listed below:
>
> # 1.	Mesh Storage
>
> We appreciate the reviewer’s insightful comment. In practice, when extracting meshes from optimized Gaussians, we have considered such issue, and have optimized the workflow to reduce storage costs. Specifically, planar priors are extracted and stored in a compact point-cloud format early in the pipeline, enabling us to refine the mesh directly without requiring storage of the intermediate default mesh (the mesh extracted by baseline methods). The default mesh extraction step is retained as an optional process for scenarios where intermediate results are needed.
>
> # 2.	TNT dataset
>
> Here we would like to state the specific metrics for each scene in TNT dataset.
> | Scene          | 3DGS | +ours | 2DGS | +ours | GOF | +ours | RaDe-GS | +ours | PGSR | +ours |
> |----------------|--------|-------|-------|-------|------|-------|--------|-------|-------|-------|
> | Barn           | 0.13   | 0.29  | 0.41  | 0.44  | 0.51 | 0.57  | 0.43   | 0.48  | 0.66  | 0.68  |
> | Caterpillar    | 0.08   | 0.10  | 0.23  | 0.23  | 0.41 | 0.40  | 0.32   | 0.32  | 0.44  | 0.43  |
> | Courthouse     | 0.09   | 0.21  | 0.16  | 0.19  | 0.28 | 0.29  | 0.21   | 0.27  | 0.20  | 0.23  |
> | Ignatius       | 0.04   | 0.08  | 0.51  | 0.51  | 0.68 | 0.67  | 0.69   | 0.68  | 0.81  | 0.80  |
> | Meetingroom    | 0.01   | 0.13  | 0.17  | 0.24  | 0.28 | 0.31  | 0.25   | 0.31  | 0.33  | 0.35  |
> | Truck          | 0.19   | 0.22  | 0.45  | 0.47  | 0.59 | 0.60  | 0.51   | 0.53  | 0.66  | 0.66  |
> | **Mean**       | 0.09   | 0.17  | 0.32  | 0.34  | 0.46 | 0.47  | 0.40   | 0.42  | 0.52  | 0.52  |
>
> From the metrics detail of TNT, we can see that in “Barn”, “Courthouse”, “Meetingroom”, our method can bring promising improvements on most of the baseline methods. This is because that in these scenes, there are more visible planes for processing. In “Caterpillar” and “Ignatius”, there appears to present less planes on foreground object, so the performance is limited.
>
> We would like to further clarify the limitations of the TNT dataset in fully reflecting the advantages of our method. While the TNT dataset serves as the only outdoor dataset with geometry ground truth (which is necessary to conduct experiments on), we believe its evaluation metrics should not be treated as the sole judgment of our method for the following reasons:
>
> **(a)**
> The ground truth in the TNT dataset consists solely of pointmap distributions, which lack explicit connectivity or topological information. As a result, the metrics provided by TNT primarily reflect vertex-wise deviations of the reconstructed mesh, rather than the quality of planar representation or surface topology.
>
> **(b)**
> Unlike ScanNet, where planar information ground truth can be obtained via mesh and InstanceID (like PlanarRecon), TNT does not provide sufficient planar-related annotations. This absence makes it challenging to evaluate the planar reconstruction performance of our method, which is one of its main contributions.
>
> Our method is specifically designed to deliver accurate and smooth topology and geometry on planar surfaces, an aspect that is difficult to directly evaluate under the current TNT metrics. While we report results on TNT to ensure comparability with prior works, we believe the reported metrics on this dataset should only serve as a reference, as they are not fully representative of the strengths of our method.

---

> ### Author Response · Authors · 2025-11-21
>
> # 3.	Monocular normal estimation for SOTA methods
>
> We further conduct monocular normal supervision on the ‘rendered_normal’ of PGSR following the same setting, both presented below and updated in Fig.5 of PDF:
> | Method              | Acc     | Comp    | Prec    | Recall  | F-score |
> |---------------------|---------|---------|---------|---------|---------|
> | PGSR                | 0.0791  | 0.0852  | 0.5815  | 0.5711  | 0.5733  |
> | PGSR+normal         | 0.0732  | 0.0710  | 0.6124  | 0.6095  | 0.6109  |
> | PGSR+ours-train     | **0.0654**  | **0.0633**  | **0.6332**  | **0.6401**  | **0.6345**  |
>
> According to the table, in both 2DGS and PGSR, our structured planar representation outperforms results with direct monocular normal supervision, while in RaDe-GS it shows comparable. This is because that RaDe-GS are less stable when encountering low resolution samples from ScanNet, resulting in poor geometry (compared with 2DGS and PGSR in Fig.3). Considering our structured planar representation as one of the key contributions of our work, it is not merely designed for final metrics improvements, but is also indispensable for our mesh layout refinement and supportive plane correction. In light of these points, we believe that directly judging the advantages of our structured planar representation solely based on the table in Fig.5 may overlook its significance. While the performance metrics demonstrate the competitive nature of our method, the structured planar representation provides irreplaceable functionality that monocular normal supervision fundamentally cannot achieve.
>
> # 4.	Efficiency
>
>  Planar Prior Processing | SAM (per img)      | Metric3D (per img) | 2D prior (per img) | 3D prior  |
> |--------------|--------------------|--------------------|--------------------|-----------|
> | Planar Prior Processing   | 14.72s             | 4.96s              | 2.78s              | 182.45s   |
>
>  Training | Initialization      | Densify | Training | Save Ply |
> |--------------|--------------------|--------------------|--------------------|-----------|
> |  Training  | +5.91s           | +2.80s              | +473.68s             | +2.04s   |
>
>  Mesh Extract| 2DGS | GOF(Tetra)| PGSR| GSPlane |
> |--------------|--------------------|--------------------|--------------------|-----------|
> |  Mesh Extract| 43.52s           | 8 * 494.17s             | 71.49s             | +37.71s   |
>
> We measured runtime overhead for each step of our method and compared it to baselines. Results are conducted on a single A100-40GB with the average performance on ScanNet dataset. The added time in mesh extraction phase of GSPlane is the average additional time cost for doing refinement. The results are also updated in Section E of Appendix.
>
> # 5.	DGR Performance
>
> The design of DGR was motivated by the potential limitations and inaccuracies introduced by off-the-shelf models in providing planar prior knowledge, which can misclassify certain Gaussians as planar. While the performance boost from DGR is indeed modest, this aligns with our expectation, as the off-the-shelf models generally provide reasonably accurate planar priors. Nevertheless, DGR plays a critical role in dynamically detecting and addressing those rare but impactful misclassifications during training, ensuring robust performance in challenging scenarios. Its contribution lies not in achieving a substantial performance gain but in safeguarding against potential errors that could degrade the overall performance. This demonstrates the necessity of DGR as a complementary mechanism, even when working with high-quality priors.
>
> For the hyperparameters of DGR, we have further conducted ablation studies on the value of 5% and 20%, results are given both below and in the updated manuscript:

---

> ### Author Response · Authors · 2025-11-21
>
> | Top % planar (with non-planar @ 20%) | Prec ↑ | Recall ↑ | F-score ↑ | Final Planar Ratio |
> |--------------------------------------|--------|----------|-----------|--------------------------------------------|
> | 1%                                   | 0.649  | 0.618    | 0.632     | 0.19                                       |
> | 2%                                   | **0.650**  | 0.616    | 0.632     | 0.19                                       |
> | 5%                                   | **0.650**  | **0.620**    | **0.633**     | 0.17                                       |
> | 10%                                  | 0.648  | 0.615    | 0.631     | 0.13                                       |
>
> | Top % non-planar (with planar @ 5%) | Prec ↑ | Recall ↑ | F-score ↑ | Final Planar Ratio |
> |--------------------------------------|--------|----------|-----------|--------------------------------------------|
> | 5%                                   | 0.649  | 0.617    | 0.633     | 0.18                                       |
> | 10%                                   | 0.651  | **0.620**    | 0.633     | 0.18                                       |
> | 15%                                   | **0.653**  | 0.618    | **0.635**     | 0.18                                       |
> | 20%                                  | 0.650  | **0.620**    | 0.633     | 0.17                                       |
> | 30%                                  | 0.647  | 0.616    | 0.632     | 0.15                                       |
> *Above are results conducted on 2DGS baseline with ScanNet samples.*
>
> | Top % planar (with non-planar @ 20%) | Prec ↑ | Recall ↑ | F-score ↑ | Final Planar Ratio |
> |--------------------------------------|--------|----------|-----------|--------------------------------------------|
> | 1%                                   | 0.631  | 0.637    | 0.632     | 0.19                                       |
> | 2%                                   | **0.633**  | 0.638    | 0.633     | 0.18                                       |
> | 5%                                   | **0.633**  | **0.640**    | **0.634**     | 0.18                                       |
> | 10%                                  | 0.632  | 0.637    | 0.633     | 0.15                                       |
>
> | Top % non-planar (with planar @ 5%) | Prec ↑ | Recall ↑ | F-score ↑ | Final Planar Ratio |
> |-------------------------------|--------|----------|-----------|--------------------------------------------|
> | 5%                                   | 0.629  | 0.635    | 0.630     | 0.21                                       |
> | 10%                                   | 0.631  | 0.637    | 0.632     | 0.19                                       |
> | 15%                                   | 0.631  | 0.639    | 0.633    | 0.18                                       |
> | 20%                                  | **0.633**  | **0.640**    | **0.634**     | 0.18                                       |
> | 30%                                  | 0.628  | 0.631   | 0.630     | 0.14                                       |
> *Above are results conducted on PGSR baseline with ScanNet samples.*
>
> The results listed in the table should be compared with the setting of ‘+ Train’ in the left table of Fig.5. From the tables it can tell that 5% and 20% yields promising results among different settings, and show its robustness across different baselines. Results are included in the main body of the manuscript in Fig.6.

---

### Official Review · Reviewer_NY5y · 2025-11-02

**Soundness:** 3
**Presentation:** 2
**Contribution:** 2
**Rating:** 4
**Confidence:** 4

**Summary:**

The paper proposes GSPlane, a 3D reconstruction method that enhances the geometric accuracy and structural simplicity of reconstructed meshes, particularly in planar regions (e.g., floors, walls, tables). The main idea is to take plane detections from 2D methods to serve as priors in 3DGS. They also propose strategies such as a dynamic Gaussian re-classifier and supportive plane correction to further refine the results.

**Strengths:**

Reparameterizes Gaussian coordinates using basis points on detected planes by 2D methods, ensuring Gaussians adhere to planar constraints during training.
Identifies and corrects misclassified planar Gaussians during training to improve planar integrity  (DGR).
An application that preserves planar integrity when decoupling objects (SPC).

**Weaknesses:**

Very few visualizations. In all figures, I cannot find comparisons that presents in the same modality. For PGSR in the teaser, it present rendered image, but GSPlane presents rendered geometry. In Figure 3, other methods presents rendered geometry while +Ours presents rendered normal maps. I cannot tell the improvements.

The accuracy of planar detection relies heavily on off-the-shelf models (e.g., SAM, Metric3Dv2). Errors in these models can propagate into the 3D reconstruction.

The paper does not address how to extend the structured representation to non-planar surfaces, which remains an open and bigger challenge for detailed geometry reconstruction.

The VGGT model gives a very smooth plane reconstruction, which is not compared.

Overall, I cannot find a significant performance improvement over other methods in the current version.

**Questions:**

See above.

---

> ### Author Response · Authors · 2025-11-21
>
> Thank you for pointing out the concerns and questions, we are grateful for your hard work on reviewing our manuscript. The responses to the main concerns are as follows.
>
> # 1.	Visualization confusion
>
> Thank you for pointing out the issue with the inconsistency in our visualizations. We have carefully revised the visualizations to ensure a more consistent and direct comparison.
>
> In the teaser, we have updated the presentation to ensure that both PGSR and GSPlane are shown in the same modality, using rendered geometry for both. For Figure 3, we have re-arranged the visualizations to present side-by-side comparisons in the same modality. Specifically, for each sample, the left side now shows the wireframe of reconstructed raw geometry, while the right side shows the wireframe of the refined geometry obtained with our method. Additionally, we have included two more samples for better comparisons to improve clarity.
>
> We hope these revisions address the concerns about visualization consistency. If there are any further issues, please let us know.
>
> # 2.	Off-the-shelf models
>
> We thank the reviewer for pointing out the reliance of planar detection accuracy on off-the-shelf models (e.g., SAM, Metric3Dv2) and the potential accumulation of their errors into 3D reconstruction. Our DGR module is proposed to address such concern by dynamically identifying and rectifying potential misclassifications of planar Gaussians during training. Importantly, the results of our ablation study (left table in Fig.5) show that DGR has only a minor impact on evaluation metrics, which further demonstrates the reliability of the off-the-shelf models in providing accurate planar priors. However, DGR remains a crucial component to handle those rare but impactful errors, ensuring robustness in scenarios where misclassifications might otherwise degrade performance. For more details about the ablation about DGR and its hyperparameters, please refer to Fig.6 in the updated manuscript.
>
> # 3.	Non-planar areas
>
> We thank the reviewer for this insightful comment. Our work focuses on planar surfaces, as planar structures are fundamental and widely present in man-made environments, making them a critical starting point for structured geometry reconstruction. We acknowledge that extending structured representations to non-planar surfaces is indeed a significant and challenging problem in detailed geometry reconstruction. While this is outside the current scope of our work, discussing non-planar refinement is still meaningful, as it aligns closely with our envisioned future work and ongoing research directions. In particular, we foresee two possible avenues to address this challenge:
>
> **(1) Extending the structured representation to more complex geometric primitives:**
> Beyond planar surfaces, structured representations could incorporate additional geometric elements such as circular arcs, curved surfaces, or other parametric shapes. These primitives could be combined with transformations like translations or rotations to represent more complex geometries.
>
> **(2)	Modeling non-planar surfaces as locally planar approximations:**
> nother potential approach involves representing non-planar regions through a dense arrangement of micro-planar patches. By reducing the scale of the structured planar representation, it is possible to approximate non-planar surfaces with a higher degree of fidelity.
>
> We believe that these directions offer promising opportunities to generalize our method and address the broader challenge of reconstructing highly detailed and non-planar geometries. While these ideas are outside the current scope of this work, they will be a key focus of our future research. We thank the reviewer again for bringing attention to this critical challenge and inspiring further exploration in this area.

---

> ### Author Response · Authors · 2025-11-21
>
> # 4.	VGGT model
>
> We appreciate the reviewer’s comment regarding the VGGT model. VGGT faces scalability limitations when handling large amount of samples such as ScanNet and TNT due to its inability to efficiently process large image sequences in a single pass, often resulting in out-of-memory (OOM) issues. This makes VGGT less suitable for processing reconstruction with long image sequences.
>
> Furthermore, the VGGT model produces a pointmap as its reconstruction output, consisting of discrete points without topological or structural information. It does not inherently generate smooth plane reconstructions. Additionally, VGGT operates in a camera pose-free manner, which fundamentally differs from GS-based methods like ours, making direct comparisons less straightforward.
>
> That said, we have conducted experiments to compare VGGT’s reconstruction results with ours as per the reviewer’s suggestion. The visualization of these results has been included in the appendix Section F. As shown, VGGT reconstructions produce only colored discrete pointmaps for visualization, which lack any topological information about the relationships between points. Consequently, the concept of a “smooth plane” is not applicable to VGGT’s outputs. We kindly invite the reviewer to refer to the appendix for further details.

---

### Author Response · Authors · 2025-11-27

Thanks again for your insightful comments and valuable time devoted to our paper. As the author-reviewer discussion period is coming to an end, please let us know if there's any further information we can provide to facilitate the discussion process. We are happy to answer any questions or concerns you may have during the author-reviewer discussion period.

---

### Meta-Review · Area_Chair_ckNo · 2025-12-30

**Summary:**

In the paper, the authors proposed GSPlane, which improves planar surface reconstruction in GSS–based methods by introducing explicit planar priors derived from SAM mask proposals and Metric3Dv2 surface normals. The method clusters Gaussians into plane groups and enforces a structured plane parameterization to enhance planar consistency. It further introduces a Dynamic Gaussian Re-classifier (DGR) to identify and revert likely false-positive planar Gaussians back to unconstrained xyz optimization, mitigating over-regularization. In addition, a mesh layout refinement step replaces dense and irregular planar mesh interiors with a regular planar grid and triangulation, significantly reducing the number of planar vertices while improving mesh topology. An optional application, Supportive Plane Correction (SPC), focuses on object–support-plane contact regions, enabling object decoupling while preserving the integrity of the supporting surface.

The main strength of the proposed method lies in its clear and consistent improvements in planar geometry reconstruction and refinement, with thorough comparisons against multiple GS baselines. However, the approach relies heavily on external priors, offers limited support for non-planar geometries, and exhibits relatively weak performance on the TNT benchmark. Overall, although the authors have made efforts to address the key concerns, the responses are not sufficiently convincing, and the evaluation scores remain unchanged. Therefore, I recommend rejection.

**Reviewer Concerns:**

Initially, the reviewers raised several recurring concerns: 1) reliance on off-the-shelf planar priors (SAM/Metric3Dv2) and potential error propagation, 2) evaluation clarity and fairness (e.g., inconsistent visualization modalities and GOF meshing comparability), 3) limited or unclear improvements (especially on TNT), 4) heuristic hyperparameters in DGR, and 5) scope limitations, since the structured representation targets planes and does not extend to non-planar surfaces.

In the end, the main issues that remain open are: 1) the approach is still fundamentally bounded by the quality and domain robustness of external priors, 2) extending the structured representation beyond planes is explicitly left as future work, and 3) some benchmarks are not convincing.

**Reviewer Scores:**

The reviewers did not engage much in the rebuttal and keep their score unchanged.

---

### Decision · Program_Chairs · 2026-01-26

Reject